# MES-CTNet: A Novel Capsule Transformer Network Base on a Multi-Domain Feature Map for Electroencephalogram-Based Emotion Recognition

**DOI:** 10.3390/brainsci14040344

**Published:** 2024-03-30

**Authors:** Yuxiao Du, Han Ding, Min Wu, Feng Chen, Ziman Cai

**Affiliations:** 1School of Automation, Guangdong University of Technology, Guangzhou 510006, China; yuxiaodu@gdut.edu.cn (Y.D.); 2112204218@mail2.gdut.edu.cn (H.D.); 2112304309@mail2.gdut.edu.cn (Z.C.); 2School of Automation, China University of Geosciences, Wuhan 430074, China; wumin@cug.edu.cn

**Keywords:** electroencephalogram, emotion recognition, multi-domain feature, capsule neural network, Transformer

## Abstract

Emotion recognition using the electroencephalogram (EEG) has garnered significant attention within the realm of human–computer interaction due to the wealth of genuine emotional data stored in EEG signals. However, traditional emotion recognition methods are deficient in mining the connection between multi-domain features and fitting their advantages. In this paper, we propose a novel capsule Transformer network based on a multi-domain feature for EEG-based emotion recognition, referred to as MES-CTNet. The model’s core consists of a multichannel capsule neural network(CapsNet) embedded with ECA (Efficient Channel Attention) and SE (Squeeze and Excitation) blocks and a Transformer-based temporal coding layer. Firstly, a multi-domain feature map is constructed by combining the space–frequency–time characteristics of the multi-domain features as inputs to the model. Then, the local emotion features are extracted from the multi-domain feature maps by the improved CapsNet. Finally, the Transformer-based temporal coding layer is utilized to globally perceive the emotion feature information of the continuous time slices to obtain a final emotion state. The paper fully experimented on two standard datasets with different emotion labels, the DEAP and SEED datasets. On the DEAP dataset, MES-CTNet achieved an average accuracy of 98.31% in the valence dimension and 98.28% in the arousal dimension; it achieved 94.91% for the cross-session task on the SEED dataset, demonstrating superior performance compared to traditional EEG emotion recognition methods. The MES-CTNet method, utilizing a multi-domain feature map as proposed herein, offers a broader observation perspective for EEG-based emotion recognition. It significantly enhances the classification recognition rate, thereby holding considerable theoretical and practical value in the EEG emotion recognition domain.

## 1. Introduction

Emotion can reflect a person’s immediate state [1]. Emotion recognition has already displayed importance in brain–computer interfaces(BCIs) [2,3], psychological state assessments [4], driver state detection [5], and cerebral workload assessments [6]. Emotion recognition can be classified into two broad types according to the signal category: non-physiological signals, such as facial expression [7], voice tone, etc. [8], and physiological signals, such as those captured on an EEG [9], ECG [10], EDA [11,12], etc. With non-physiological signals, although they express emotions more intuitively, it is easier for a person to mask their true state according to their subjective wishes. In contrast, physiological signals can objectively express the truest emotional states [13]. Many neuroscience research studies have shown that EEG signals are more intimately associated with emotions than ECG signals are [14], so emotion recognition based on brainwaves shows more accurate results.

In the past, researchers explored how to accurately and effectively recognize emotional information in EEG signals. In earlier studies, researchers were more likely to use machine learning (ML) methods whose main approach was to manually extract features as inputs, such as support vector machines (SVMs), KNN [15], random forests [16], decision trees, etc., or a model that integrates multiple classifiers [17]. Among these methods, the ML algorithm mainly uses some of the more popular EEG signal-based features, such as Higher-Order Crossing (HOC) [18], Hjorth features [19], Power Spectral Density (PSD) [20], Differential Entropy (DE) [21], Sample Entropy (SE) [22], and Differential Asymmetry (DASM).

In recent years, as deep learning (DL) algorithms have become increasingly optimized and computer performance has increased dramatically, brainwave emotion recognition models using deep learning have been effectively applied and demonstrated excellent performance. Thammasan et al. proposed the use of Deep Belief Networks (DBN) [23] to classify emotion from manually extracted electroencephalographic power spectra (PSD) and discrete wavelet features. Algarni et al. proposed a stacked bi-directional LSTM (Bi-LSTM) model [24] and classified it after feature selection using the gray wolf algorithm. Song et al. designed a dynamical graph convolutional neural network (DGCNN) model [25] which is able to extract feature and functional connectivity information simultaneously. Convolutional neural networks (CNNs), as classical network structures in deep learning, show excellent performance in brainwave-based emotion recognition research, and Gao et al. proposed a multilayer convolutional neural network (MCNN) [26] based on combining both differential entropy (DE) and a brain network for emotion classification. Some researchers use a hybrid model approach for emotion classification; Zhang [27] et al. proposed an end-to-end hybrid model that focuses on combining CNNs with LSTMs and using raw EEG signals as input for an emotion-recognition model.

However, existing methods still have certain limitations and areas for improvement. Firstly, as mentioned earlier, raw EEG signals carry a great deal of information. The features of different domains, the features of different frequency bands, the method of placing the signal acquisition device, temporal relationships between consecutive EEG signals, etc., can convey unique emotional information. Unfortunately, previous studies mainly focused on a single feature dimension, and the observation horizon of EEG signals needs to be more comprehensive. The lack of data structures that can comprehensively cover the characteristics of the multi-domain features of EEG waves as inputs into a recognition model may cause the model to ignore key emotional information.

Although DL methods are a significant improvement over traditional methods, most DL methods are improved via CNNs. The spatial characteristics of feature information and emotional state are closely connected in brainwave signals with multi-domain features. Still, the CNN network focuses more on the probability of a certain feature’s existence in the feature extraction and recognition process. Ignoring spatial distribution information between features is a disadvantage.

Aiming to solve these existing problems, we propose a new EEG emotion recognition model based on multi-domain feature fusion named MES-CTNet, which has a model framework mainly comprising an improved multichannel CapsNet and Transformer-based temporal coding layers. To focus on more levels of information in the EEG signal, this study proposes a multi-domain feature map for multi-domain feature reconstruction as an input to the model. Firstly, the original EEG signals are preprocessed. The signals in the four frequency bands of *α*, *β*, *γ*, and *θ* are extracted, and the signals in different frequency domains are characterized by their differential entropy (DE), sample entropy (SE), and power spectrum density (PSD); finally, the preprocessed features are spatially mapped according to the spatial distribution of electrodes. The multi-domain feature map is finally formed as an input to the model. The multi-domain feature map fuses the space–frequency–time properties of multi-domain features through a simple structure, allowing it to better mine multi-level information about emotions in EEG signals. Next, a multichannel CapsNet performs multi-domain feature fusion and extracts multi-domain feature maps. Compared with CNNs, the ability of the multichannel CapsNet [28] to focus on the spatial distribution of features can fully explore the emotional information embedded in multi-domain feature maps. To enhance the ability of the CapsNet to extract effective spatial features and suppress invalid features, ECA blocks [29] were added between the first convolutional layer of the CapsNet and the primary capsule, and SE blocks [30] were embedded inside the primary capsule to optimize model performance without making the model more complex. The improved multichannel neural network obtains emotion representation vectors from current time slices. Transformer has a strong global sensing ability [31] to tap into the properties of EEG signals in the time domain; the emotion representation vector of continuous time slices is used as semantic information on emotional state, and the temporal coding layer of the Transformer is utilized for global coding. Finally, the softmax classifier predicts outputs with emotion labels.

The main contributions of this paper are as follows:We propose a new multi-feature fusion-based emotion recognition model called MES-CTNet which innovatively combines the spatial perception capability of CapsNet and the global semantic analysis capability of Transformer to fully exploit the multi-domain feature information of EEG signals. We embed ECA blocks and SE blocks into CapsNet to improve the effectiveness of the direct transfer of feature information in capsules without increasing the complexity of the model and achieve more effective recognition.We construct a new multi-domain feature map which utilizes a simple structure to fuse the spatial–frequency band–temporal characteristics of the multi-domain features of the EEG signals, making the model focus more comprehensively on the multi-level feature information and providing data-level support for the model’s recognition to achieve higher recognition accuracy.The method demonstrates excellent performance on the globally recognized sentiment recognition datasets DEAP and SEED, with average accuracies of 98.31% and 98.28% for classification performed on DEAP’s valence and arousal. The average accuracy of triple classification on the SEED dataset reached 94.91%.

The sections of this paper are as follows: Section 2 describes the materials and [material and methods used in our work, including the construction of the multi-domain feature map and the structure of MES-CTNet. The experimental results are provided in Section 3, and the experimental result are discussed in Section 4. In Section 5, we conclude our research work.

## 2. Materials and Methods

This paper proposes a novel capsule Transformer network based on the multi-domain feature for EEG-based emotion recognition, referred to as MES-CTNet, and Figure 1 illustrates the MES-CTNet framework. It mainly includes EEG signal acquisition, time slice segmentation and preprocessing, multi-domain feature computation, multi-domain feature map construction, an improved multichannel CapsNet part, a Transformer-based temporal coding layer, and a Softmax classifier part.

Firstly, we reconstructed an original EEG signal, extracted the EEG’s features from multiple domains, and constructed a multi-domain feature map by combining the spatial, band, and temporal characteristics of the electrodes and then fused and extracted the features within the slices of the multi-domain feature map using a multichannel CapsNet embedded in the ECA and SE blocks; finally, we re-encoded the feature information of multiple consecutive slices through the Transformer-based temporal coding layer to explore feature connections between slices. After that, the feature information of multiple straight-time slices was recorded by the Transformer-based temporal coding layer to mine the feature links between slices, and finally, the final emotional state was obtained by the Softmax classifier.

The following are specific descriptions of each component.

### 2.1. Descriptions of Databases

#### 2.1.1. DEAP Dataset

The DEAP dataset [32] collected EEG data from 32 healthy people (16 males and 16 females; mean age: 26.9) in different emotional states induced by 40 music videos lasting 60 s. The experiment was conducted using 32-lead electrode caps attached according the “10–20” system on each participant’s frontal, parietal, occipital, and temporal lobes of the brain with a sampling frequency of 512 Hz. The participants were asked to rate the videos as they watched them on a scale of 1–9 based on emotional state using a form representing “Valence” and “Arousal” to provide immediate emotional state markers. According to the valence–arousal scale mentioned by Russell [33], we take valence and arousal, the most commonly used dimensions, as labels for categorization and use the median value of 5 as the threshold value to classify valence and arousal to perform a binary categorization task for emotion recognition. The experimental acquisition of the DEAP dataset is presented in Figure 2.

#### 2.1.2. SEED Dataset

The SEED dataset [21,34] captured the emotional state EEG data of 15 Chinese participants (7 males and 8 females; mean age: 23.27) stimulated by 15 Chinese movie clip frames lasting approximately 4 min; the video clips can be categorized into 3 types of emotional labels (positive, neutral, and negative), with each clip represented only one emotional label. While watching the video, the researcher captured brain waves from each subject’s brain region at a frequency of 1000 Hz using a 62-lead electrode cap with standard leads and the “10–20” system. The above participants were subjected to the above experiment three times, each time with a one-week interval, and a total of 45 sets of EEG data were obtained. The experimental acquisition of the SEED dataset is presented in Figure 3.

### 2.2. Feature Extraction Methods

Several features are extracted in this paper to obtain richer and more effective emotional information on EEG signals, including the differential DE, SE and PSD. The DE [35,36] feature is the most reliable feature, reflecting the emotional changes in previous studies. As a nonlinear dynamics-based feature type, the SE feature [37] is well suited for studying EEG signals. PSD [38,39,40] features are extensively applied to study the power distribution of EEGs and provide effective value for EEG-based emotion recognition.

The DE feature can be expressed by Equation (Equation 1).
(1)$$\begin{array}{ccc} DE(X) & = & \frac{1}{2}\log\left(2\pi e\sigma^{2}\right) \end{array}$$

Here, $\sigma^{2}$ denotes the variance in the time series EEG signal to be extracted.

The SE feature can be expressed by Equation (Equation 2).
(2)$$\begin{array}{c} SampEn\left(m,r\right)=-\;ln\frac{B^{m+1}\left(r\right)}{B^{m}\left(r\right)} \end{array}$$

$B^{m}\left(r\right)$ is the probability that two sequences match *m* points under a similarity tolerance *r*. $B^{m+1}\left(r\right)$ is the same.

The steps for extracting the PSD feature are as described below:

First, a discrete Fourier transform is applied to the discontinuous EEG signal u(n)(n=0,1,⋯,N−1), as shown in Equation (Equation 3).
(3)$$\begin{array}{c} U\left(k\right)=\sum_{n=0}^{N-1}\;u\left(n\right)W_{N}^{kn} \end{array}$$
where $W_{N}=e^{-j2\pi /N},k=0,1,\cdots ,N-1\;.$ Since $W_{N}$ is periodic, U(k) is also periodic, with *a* period size of *N*. The PSD of the EEG signal can be found using Equation (Equation 4).
(4)$$\begin{array}{c} PSD=U^{2}\left(k\right)/N \end{array}$$

### 2.3. Multi-Domain Feature Map

#### 2.3.1. Data Preprocessing

We preprocessed the EEG data from both datasets by downsampling the DEAP dataset to 128 Hz and the SEED data to 200 Hz according to methods from previous research. In the DEAP dataset, we sliced the signal using a non-overlapping sliding window of 0.5 s, yielding 4800 samples per individual. We used a 2 s recognition cycle, i.e., one sample per 4 time slices, resulting in 1200 samples for EEG emotion detection. Similarly, in the SEED dataset, we partitioned the signal using a non-overlapping sliding window of a 1 s duration, fusing the three time slices to obtain 10,152 samples per individual. With 4 time slices as a sample of the recognition cycle, 2538 samples of EEG emotion detection were obtained.

#### 2.3.2. Multi-Domain Feature Map Construction

The feature information of different domains reflects the state of brainwaves from different observation perspectives. To fully utilize the expressive ability of EEG signals for emotions with a simple data structure, we constructed a new multi-domain feature map structure that can combine the important spacial–frequential–temporal properties of multi-domain features.

In EEG-related research, signals in different frequency band ranges are often separated for processing according to different research needs. In related research, signals in four different frequency bands, *θ* (4–8 Hz), *α* (8–13 Hz), *β* (13–31 Hz), and *γ* (31–50 Hz), are closely related to emotional states [41]. Therefore, to improve the quality of EEG data on emotional information, a Butterworth filter of order 3 and type bandpass is used to extract the four frequencies described above as base data.

EEG signal capture systems typically place electrodes according to a “10–20” system; therefore, to maintain the spatial characteristics of EEG signals between different channels, after extracting the above features, similar to previous work [42,43,44,45], we mapped various types of features in the spatial matrix based on the distribution of electrodes used in the collection of the dataset, and Figure 4 demonstrates that the frequency bands of the various types of features are mapped onto an 8 × 9 matrix according to two different international standards.

To give full play to the ability to characterize emotions in different features of EEG signals, this paper fused the above features with different characteristics. Firstly, the raw EEG signals were divided into four bands of *α*, *β*, *γ*, and *θ* frequencies, and θ were obtained. Next, the signals in each frequency band were sliced using a non-overlapping window.Then, the DE, SE, and PSD features were computed according to each sliced signal’s channels. After that, each temporal slice was mapped according to the frequency band distribution and the multivariate feature information according to the spatial distribution of the electrode placement, resulting in an 8 × 9 matrix. Finally, we adopted a 3-dimensional splicing method to splice the feature matrices of all frequency bands to form a 16 × 18 feature matrix. Then, three 16 × 18 matrices with different features were fused and constructed in parallel to create a 3 × 16 × 18 multidomain feature map which integrates the spatial characteristics, frequency band characteristics, and multi-dimensional feature characteristics of the EEG signals. Figure 5 shows the process of constructing a multidomain feature map with 32 channels as an example. The total length of the experimental data by data preprocessing and construction is given in Table 1.

### 2.4. MES-CTNet for EEG Base Emotion Recognition

The core of MES-CTNet consists of a multichannel CapsNet embedded with ECA and SE blocks cascaded with a Transformer-based temporal coding layer. Among them, the CapsNet is responsible for intra-slice feature fusion and the advancement of multi-domain feature maps. Transformer is responsible for re-encoding feature information between slices.

#### 2.4.1. ECA Block

ECA (Efficient Channel Attention) [29] is a channel attention module widely used in visual perception modeling. It can perform channel feature enhancement on multichannel input features [46], and the size of the feature map will not be altered after enhancement using ECA. Hence, the ECA module can enhance the model without increasing the complexity of modeling benefits.

Figure 6 describes the structure of the ECA block; firstly, during capsule network delivery, the multichannel feature information is compressed into a 1 × 1 × *C* structure by global average pooling, aggregating all the features. *k* can be connected to the *C* channel and adjusted for *C*, as shown in Equation (Equation 5).
(5)$$\begin{array}{c} k=\psi \left(C\right)=|\frac{log_{2}\left(C\right)}{\gamma }+\frac{b}{\gamma }{|}_{odd} \end{array}$$

The γ and *b* represent the offsets of linear mapping. Finally, a sigmoid is used to generate weight ratios for each channel which the original input is multiplied by to produce an input with the channel attention property.

#### 2.4.2. SE Block

SE (Squeeze and Excitation) [30] is likewise a channel attention mechanism. It can increase the weight of valid features and eliminate the redundancy of invalid features in the feature channel to optimise the performance of an emotion perception capsule [47].

The structure of the SE block is indicated in Figure 7; it is mainly composed of two parts, Squeeze and Excitation. In the Squeeze layer, a feature block with spatial dimensions of H×W is compressed into a global description of features *Z* of a size 1×1×C, as shown in Equation (Equation 6).
(6)$$\begin{array}{c} Z_{c}=F_{sq}\left(u_{c}\right)=\frac{1}{H\times W}\sum_{i=1}^{N}\sum_{j=1}^{\infty }u_{c}\left(i,j\right) \end{array}$$
where $Z_{c}$ represents the *c*th element and $u_{c}$ represents the set of features after convolution, where the convolution kernel size is *c*.

The Excitation layer is an excitation weighting of the set of channel feature real numbers derived from the Squeeze layer to characterize the correlation and importance between the channel features. The specific expression formula is shown in Equation (Equation 7).
(7)$$\begin{array}{c} s=F_{ex}\left(z,W\right)=\sigma \left(g\left(z,W\right)\right)=\sigma \left(W_{2}\delta \left(W_{1}z\right)\right) \end{array}$$
where σ represents the RELU function, $W_{1}\in R^{\frac{C}{r}C}\;,\;\;W_{2}\in R^{\frac{C}{r}}\;$, and *r* represents the dimensionality reduction multiplier.
(8)$$\begin{array}{c} \;\;\;\chi_{c}=F_{scale}\left(u_{c},s_{c}\right)=s_{c}\cdot u_{c} \end{array}$$

Finally, according to Equation (Equation 8), the weights are multiplied with the output characteristics to yield the output of the SE block.

#### 2.4.3. Intra-Slice Emotion Recognition with MES-CapsNet

We utilize a CapsNet for within-slice feature fusion and the extraction of a multi-domain feature map. Since the space distribution of features is closely related to the emotion information in the previously constructed multi-domain feature map, this paper’s traditional CapsNet [42,48] is improved to better mine the feature information about emotion in the EEG multi-domain feature map. Since the multi-domain feature map is mapped based on the signal electrode space map, there is a close connection between the features and the spatial distribution of each channel before. Inspired by multicolor-domain convolutional channels [49] in the field of vision research, we improves the CapsNet into a multichannel CapsNet for extracting the intra-slice emotional information of the EEG signal. The improved Capsnet has the following three main modules: multichannel ConvReLU, PrimaryCaps, and EmotionCaps, and the three modules are cascaded sequentially, as shown in Figure 8. Since the input data have a 3D structure, 256 3-channel 3 × 3 convolution kernels with a step size of 2 are used in the ConvReLU layer, which serves the purpose of extracting the in-slice local spatial information for all bands and all channels in the feature map. To enhance the multichannel feature information transfer, we embedded ECA blocks between the ConvReLU layer and the PrimaryCaps layer. Next is the PrimaryCaps layer, which is composed of 32 convolutional 8D capsule channels. Each of these consists of eight 3 × 3 convolutional kernels of a step size of 1. To make the feature information in the PrimaryCaps layer highly relevant and valuable to the emotional state highlighted, we embed the SE block in the PrimaryCaps layer. The vector output’s length and direction in this layer express the probability of low-level features associated with emotional state in the feature map, as well as other characteristics. The last layer is the EmotionCaps layer, where a dynamic routing mechanism unique to the CapsNet is used to connect PrimaryCaps and EmotionCaps. This mechanism can extract and maintain the relative spatial relationship between features compared to the traditional feature pooling layer. At the same time, it can enhance the information transfer between features at different levels and ensure information consistency between local and global features, which is very important for electrode-based feature map space mapping.

Since this paper performs a binary classification task on the DEAP dataset and a tertiary classification task on the SEED dataset, two and three 16D emotion-aware capsules are set up in EmotionCaps, corresponding to different emotional states. The final vector outputted from each emotion-aware capsule expresses the emotional state as characterized by the multi-domain feature map of the current time slice.

#### 2.4.4. Inter-Slice Emotion Recognition with Transformer-Based Temporal Coding Layer

Since the EEG signal is a section of continuously changing signals, the temporal characterization of the continuous signal is also essential, but in the above CapsNet, part of the work only extracts the emotional information of each time slice and can not completely characterize the complete emotional state, which is a local slice of the data. Therefore, this paper uses a Transformer-based temporal coding [50,51] layer to temporally encode the moving representation vector of the EEG signals to realize global emotional state perception and compensate for the shortcomings of the CapsNet regarding the perception of the time series.

The temporal coding layer of the Transformer is mainly composed of Multi-Head Self-Attention (MHSA) and feed-forward network (FFN) modules paired by a residual connectivity layer and layer normalization (LN). In MHSA, the attention of each head is calculated as shown in Equation (Equation 9).
(9)$$\begin{array}{c} A=\mathrm{Attention}\left(Q,K,V\right)=\;\mathrm{softmax}\left(\frac{QK^{T}}{\sqrt{d_{k}}}\right)V \end{array}$$
where *Q*, *K*, and *V* represent the vectors of query, key, and value, respectively, derived from the corresponding mapping matrices. *Q* and *K* are multiplied pointwise. Its size is limited by $d_{k}$ to prevent the gradient from being too small after Softmax and then mined with *V* to obtain the output of the layer, which is carried out so that the output is relevant to all inputs after all inputs are encoded with attention. As shown in Figure 9 and Equation (Equation 10), the MHSA comprises multiple self-attention modules. The input of each self-attention module is a “subspace” of the original input, and by focusing on different features through multiple self-attention modules and finally connecting all the outputs through a fully connected layer, an MHSA is formed which is designed to enable the model to capture richer features.
(10)$$\begin{array}{c} MHSA\left(Y\right)=FC([A_{1}\|A_{2}\left\|....\right\|A_{H}]) \end{array}$$

Next, the output of the above multi-attention mechanism is residually concatenated and normalized and fed into the fully connected feed-forward network, which is also residually concatenated and normalized to obtain the final output of the layer, which will also serve as the input to the next temporal coding layer. The above constitutes the Transformer-based temporal coding layer, and the structure is shown in Figure 10. In this process, the output sentiment representation vector from the capsule network will be used as the input for the encoder, which will output the encoded sentiment representation vector after multiple rounds of encoding.

In the Transformer-based temporal coding layer constructed in this paper, we connect six coding layers, and each coding layer consists of eight parallel self-attention-forming MHSA modules with the FFN and LN layers. Then, as shown in Equations (Equation 11) and (Equation 12), the sentiment feature vectors of multiple time slices are tiled and then sequentially connected to construct a set of sentiment feature vectors with time domain characteristics.
(11)$$\begin{array}{c} Y_{i}=flatten\left(Y_{i}\right) \end{array}$$
(12)$$\begin{array}{c} Y=[Y_{1}\|Y_{2}\|....\|Y_{T}] \end{array}$$

The final obtained *Y* will be input to the Transformer-based temporal coding layer for a global perception of emotional state as temporal semantic features characterizing the emotional state. To explore the temporal characteristics of the emotion information of continuous EEG signals, this paper takes a set of emotion feature vectors constructed from multiple consecutive time slices as the input of the temporal coding layer, and the powerful global sensing ability of the temporal coding layer encodes the time slices in a temporal sequence. Finally, the final result is obtained by classifying the emotion representation vectors obtained after the global recoding of each time slice by a Softmax classifier.

## 3. Experiments and Results

### 3.1. Experimental Platforms

We employed pyTorch 1.9.0 to build the model and conducted the experiments on an NVIDIA A30 GPU (From NVIDIA Corporation Santa Clara, CA, USA) with 16G, using the Adam optimizer; the learning rate was set to 1 × $10^{-3}$ (DEAP) and 1 × $10^{-4}$ (SEED), the batch size was 128, and the loss function adopted was CrossEntropyLoss. In each round of experiments, the comprehensive performance of the model was evaluated using a 10-fold cross-validation strategy, i.e., a single dataset was divided into ten subsets, each of which served as a test subset in turn, and the other nine subsets served as training subsets; 10 experiments were conducted in turn, and the average result of the 10 experiments was taken as the final result of that round of experiments, which was used to evaluate the model’s generalization ability.

### 3.2. Experimental Arrangements

In the DEAP dataset, we use a subject-dependent strategy to validate each individual with separate experiments. In the SEED dataset, in order to justify the generalizability of the present model, we ignored temporal differences between the multi-session experiments, fused the data from the three sessions together, and validated each individual in a separate experiment using a subject-dependent strategy. In this paper, the feasibility of this model and the degree of optimization for existing methods were verified through multiple sets of ablation experiments and comparisons with existing state-of-the-art methods. The average of the assessment metrics for each individual is used as the experimental result for that individual. The average result for each individual is used as the overall result of the model.

### 3.3. Emotion Recognition Results of MES-CTNet

We conduct experiments on the DEAP dataset and SEED dataset and obtain an average confusion matrix of all subjects in the two datasets, as shown in Figure 11. As seen in the confusion matrix, positive emotions seem to be easier to distinguish relative to negative emotions. Meanwhile, we list the training loss of the first subject in the two datasets in Figure 12, and it can be observed that during the training process, the training loss values can decrease rapidly and converge gradually to remain stable. In addition, to better analyze the performance of the model, we list the accuracy, F1-score, recall, and precision on the DEAP and SEED datasets in Table 2. Accuracy is an intuitive representation of model performance in the balanced dataset, while we utilize precision and recall to demonstrate the proportion of positive samples (positive emotions) that are identified by the model during the classification process, and since precision and recall often need to be weighed against each other, we again utilize the F1-scores to provide a comprehensive view of the model’s classification. From the table, the model achieves excellent results in all performance metrics with a balanced distribution, demonstrating the comprehensive performance capability of the model.

### 3.4. Comparison with Advanced Methods

In order to prove the effectiveness of our method, we compared the MES-CTNet model with state-of-the-art methods, and the following is a summary of the other methods:

CDCN [52]: It proposes a deep-learning framework using a channel-fused dense convolutional network which optimizes recognition performance with densely connected channels.

4D-CRNN [53]: It develops a 4D convolutional neural network and combines it with LSTM units to effectively combine the frequency–space–time properties of EEG signals.

ACRNN [54]: It proposes an attention-based convolutional recurrent neural network that applies the adaptive allocation of channel attention to the task of emotion extraction from EEG information.

Caps-LSTM [55]: It proposes an EEG-based emotion recognition model fusing a Capsnet and LSTM and a combination of channel-wise attention to augment the input signal.

Caps-EEGnet [56]: It proposes an EEG-based emotion recognition model that fuses the Capsnet and EEGnet and combines it with the ReliefF-mRMR algorithm for channel screening.

Table 3 compares the average accuracy of the MES-CTNet method with other similar methods for multiple subjects in two datasets. From the comparative data, we note the excellent performance exhibited by our model. Among the models, compared with the latest CNN-based models, CRNN and ACRNN, the model shows a significant improvement of about 4.5% on the two labels, valence and arousal, in the DEAP dataset. Compared with the latest capsule-network-based models, Caps-LSTM and Caps-EEGnet, our model showed significantly improvement by about 1–1.7%. In the SEED data, the present model also shows the best performance compared to other cross-session-based models. In a comparison of all models, the standard deviation of our model in terms of average accuracy is the smallest, which shows that our model has the highest generalization among all individuals. At the same time, this table shows that the improvement of the CapsNet in EEG emotion recognition against the CNN model is pronounced.

### 3.5. Ablation Experiment

#### 3.5.1. Attention Ablation Experiments

Since three types of attention, ECA, SE, and MHSA, are added to the MES-CTNet, we conducted a series of experiments on attention mechanisms using two datasets for the attention mechanisms to compare the effects of different attention mechanisms on the model’s performance of the emotion recognition task. The following scenarios were set according to the attention mechanisms: without any attention (W/O Attn), with only MHSA, with only MHSA and ECA block (ECA-MHSA), with only SE block and MHSA (SE-MHSA), and with all attention. In all experiments with different attentional compositions, the model takes as its input the multi-domain feature map that incorporates multiple classes of features.

Table 4 displays the average results using different attentional strategies tested on the DEAP and SEED datasets. Figure 13, Figure 14 and Figure 15 show the results of the DEAP and SEED datasets for each subject under different attention strategies, respectively. The results show that compared with the model of W/O attention, the effect of adding the MHSA mechanism is a very obvious improvement; in the DEAP dataset, after adding the MHSA attention mechanism, the overall average recognition accuracy is improved by 5.52%, and 5.75%, respectively. In the SEED dataset, the overall average accuracy is improved by 1.50%. This is because the primary role of the MHSA mechanism is to uncover the temporal correlation of the time-slice features of continuous EEG signals, which is very important for constant EEG wave signals. Therefore, we use the model with only the MHSA mechanism as the baseline model; after embedding the upper ECA block and SE block in the CapsNet, respectively, the recognition effect of the model is again improved, which is attributed to the adjustment of the distributional weight of the feature set by the ECA block in the process of spatial feature extraction as well as the prominence of practical information due to the SE block in the process of feature information transfer. Finally, we combine the advantages of the ECA block, SE block, and MHSA to construct the final model, MES-CTNet, which demonstrates significantly improved performance compared to all the models not fully focused on above.

#### 3.5.2. Feature Ablation Experiments

In this paper, we use a multi-domain feature map that incorporates three features, DE, SE, and PSD, as input to the model. In order to investigate the effects of different feature structure inputs on the model’s ability to complete the emotion recognition task, we use the complete model MES-CTNet, which contains all attentional mechanisms, as a base model and design a series of experiments based on different feature components, including the following cases: only DE features, only SE features, only PSD features, DE + SE, DE + PSD, and all features.

Table 5 demonstrates the average accuracy and standard deviation of different feature constructions under the two datasets tested. Figure 16, Figure 17 and Figure 18 show the DEAP and SEED datasets’ results under different feature constructions for each subject, respectively. From the result data, after fusing these three types of features together, a multi-domain feature map is constructed, thus combining the advantages of each feature and significantly improving accuracy. During the experimental process of the three single features, in the DEAP data, DE has the best overall performance, showing the highest accuracy rates in both the valence and arousal dimensions, which are 95.43% and 95.33%, respectively. This is followed by the SE and PSD features. In the SEED dataset, the DE feature is still the best-performing feature among the single features, with an average accuracy of 91.37%. Therefore, we use DE features as the basic features, and after fusing SE features and PSD features, the recognition performance of the model is significantly improved, which proves that the method of combining features can absorb the advantages of the features. Therefore, when fusing the three types of features to construct a multi-domain feature map as the input for the model, the model’s performance still shows a substantial improvement. From the comparison of the rest of the performance metrics of the experiments on the two datasets, it can be found that the sensitivity and stability of the model are significantly optimized. This is because multi-domain feature map integrates the advantages of various features to enhance the ability of EEG data to express emotional states. In contrast, constructing multi-domain feature data makes the feature distribution of the data more stable, improving the model’s ability to recognize emotional states.

In addition, as this experiment adopts a subject-dependent experimental strategy, it is evident from the results that the experimental results are affected by individual differences due to the differentiation of different individuals’ perceptions of emotions. In particular, in the case of a single feature as a model input, the accuracy is more affected by individual differences, withthe SE features and PSD features showing more obvious performance. When the multi-domain feature map is used as the input to the model after fusing the three types of features, the degree of difference in the recognition effect between individuals is significantly reduced, and the stability of the recognition accuracy is improved; additionally, the standard deviation of the experimental results is the lowest when the multi-domain feature map is used as the input to the model, as shown in the data examples. This is due to the ability of the multi-domain feature map to contain more EEG sentiment information and reduce differentiation between individuals.

In summary, based on the excellent performance of the emotion recognition model in this paper, excellent results can be achieved when single and multiple features are used as inputs for emotion recognition. Still, the multi-domain fusion feature maps, which incorporate multiple features’ advantages, show the most excellent and stable performance.

## 4. Discussion

From the above experimental results, it can be seen that the method proposed in this paper has obvious advantages in subject-dependent emotion recognition tasks, and its effectiveness is verified through multiple rounds of ablation experiments to demonstrate the roles played by the main components of the method. In this section, it is therefore necessary to discuss the advantages and shortcomings of the proposed method in the emotion recognition task.

The presentation of emotional states in EEG signals is closely related to the mode of EEG signal acquisition and also to the characteristics of EEG signals under different observation angles. Therefore, the proposed framework makes use of a multi-domain feature map to achieve the inclusion of the above complex multi-domain feature information in a simple structure. Such an innovative data structure provides strong data support for the subsequent model to achieve excellent emotion recognition effects, with a more accurate and stable performance compared to single features.

The MES-CTNet model utilises the powerful ability of a multichannel Capsnet to perceive the intrinsic relationship between space and features and is able to fully fuse and mine the emotional information embedded in multi-domain feature maps. The model makes use of Transformer’s excellent contextual correlation ability to recode the emotional information, connect the temporal properties between consecutive slices, and restore the emotional representational features of consecutive EEG signals. In addition, important features are screened between Capsnet feature channel deliveries through an ECA block and an SE block to enhance the effectiveness of feature delivery. Therefore, the proposed model framework can effectively target the construction of a multi-domain feature map and comprehensively extract and integrate emotional feature information, thus demonstrating advanced EEG-based emotion recognition.

However, the present model still has some shortcomings. While the current model performs excellently in experiments with independent individuals, it performs poorly in scenarios in which cross-individual assessments are conducted, with performance much lower than in scenarios with independent individuals. This may result from inconsistencies in the degrees of physiological representations of emotional states and inconsistencies in subjective perceptions of emotional states that existed before in the individuals. In the future, this research will focus more on cross-individual studies for harmonization with practical applications.

## 5. Conclusions

With the development of the human–computer interaction era, emotion recognition based on EEG has received wide attention. However, traditional emotion recognition methods are deficient in mining the connection between multi-domain features and fitting their advantages. In this paper, we propose a new emotion recognition model based on a multi-domain feature map called MES-CTNet. First, in order to fuse more EEG information with a simple structure, we construct a new multi-domain feature map as the input for the model. This structure combines the spatial–frequential–temporal characteristics of multi-domain features to provide a more comprehensive database for the model. And a multichannel CapsNet embedded with an ECA block and an SE block is used for feature fusion and the extraction of multi-domain feature maps within time slices. The CapsNet, focusing on spatial features, retains the connection between the EEG channels and the features, and the ECA block and SE block enhance the transfer of feature information within the capsule effectively. Then, the emotional features of continuous time slices are encoded by a Transformer-based temporal coding layer, and the powerful contextual correlation ability of Transformer is utilized to mine the temporal properties of the multi-domain feature profiles to obtain the final emotional state. We conduct full experiments on two generalized datasets with different emotion labels: the DEAP dataset and the SEED dataset; average accuracies of 98.31% and 98.28% were achieved for the valence and arousal dimensions of the DEAP data, respectively, and an average accuracy of 94.91% was achieved for the SEED dataset. This paper demonstrates that the present model shows the best emotion classification performance on the experimental data examples through sufficient attention ablation experiments, feature ablation experiments, and comparisons with similar state-of-the-art methods.

## Figures and Tables

**Figure 1 brainsci-14-00344-f001:**
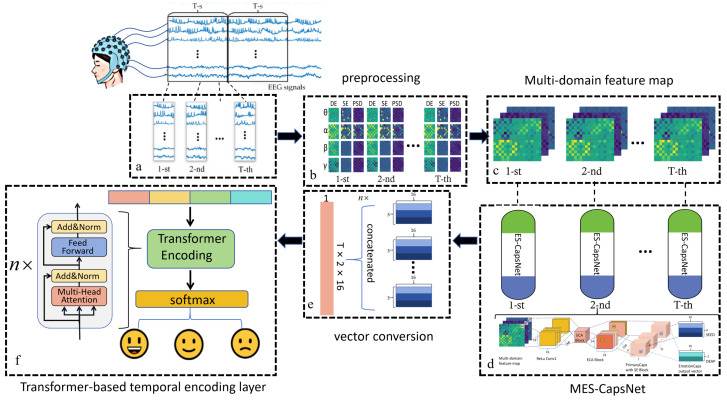
The framework diagram of a capsule Transformer Network based on multi-domain feature maps (MES-CTNet) for EEG-based emotion recognition. (**a**) EEG signal slicing; (**b**) Multi-domain feature extraction; (**c**) Multi-domain feature map construction; (**d**) Intra-Slice Emotion Recognition with MES-CapsNet; (**e**) Multi-temporal emotion representation vector connections; (**f**) Inter-Slice Emotion Recognition with Transformer-Based Temporal Coding Layer.

**Figure 2 brainsci-14-00344-f002:**
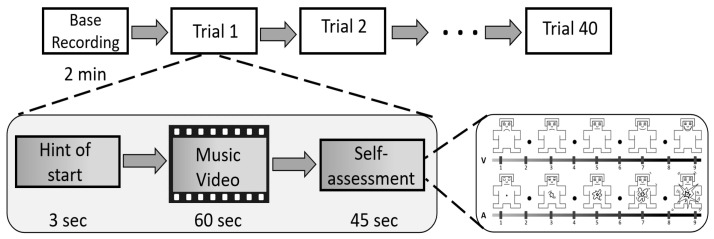
The experimental acquisition of the DEAP dataset.

**Figure 3 brainsci-14-00344-f003:**
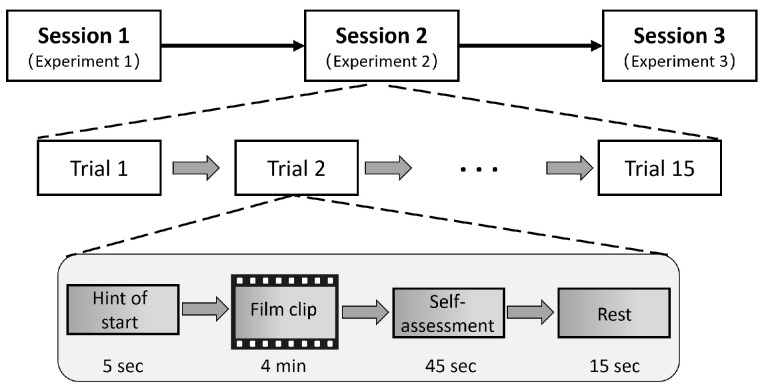
The experimental acquisition of the SEED dataset.

**Figure 4 brainsci-14-00344-f004:**
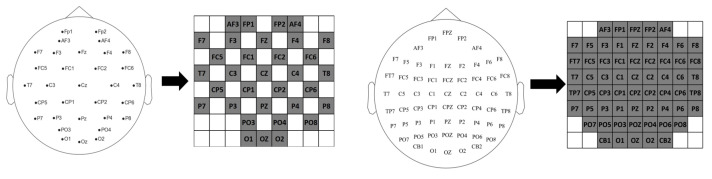
Electrode space mapping.

**Figure 5 brainsci-14-00344-f005:**
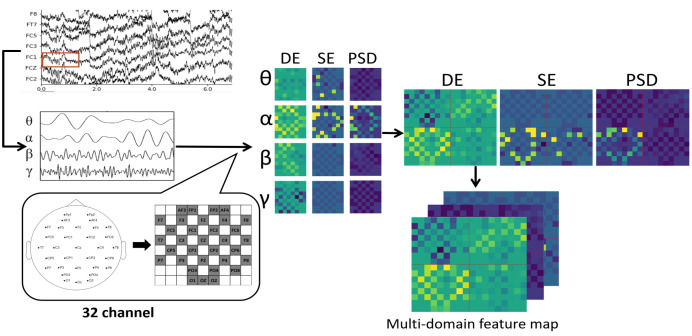
The process of multi-domain feature map construction, where the red box represent slicing the signal.

**Figure 6 brainsci-14-00344-f006:**
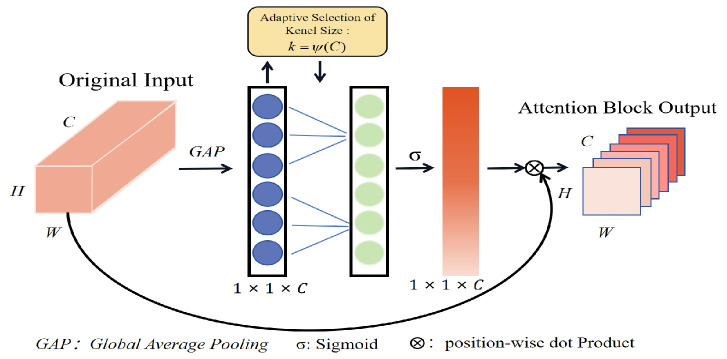
Efficient channel attention (ECA) block.

**Figure 7 brainsci-14-00344-f007:**
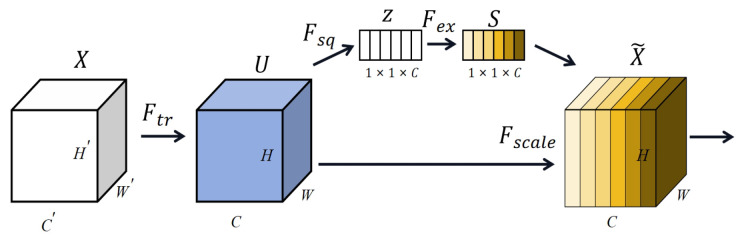
Squeezeand Excitation (SE) block.

**Figure 8 brainsci-14-00344-f008:**
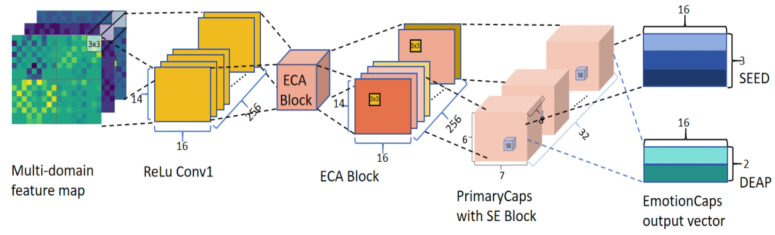
The framework of a multichannel CapsNet with ECA and SE blocks, where different colored feature channels represent different weights assigned after ECA block processing.

**Figure 9 brainsci-14-00344-f009:**
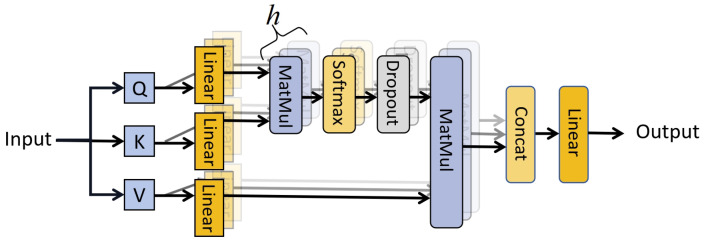
Multi-Head Self-Attention (MHSA).

**Figure 10 brainsci-14-00344-f010:**
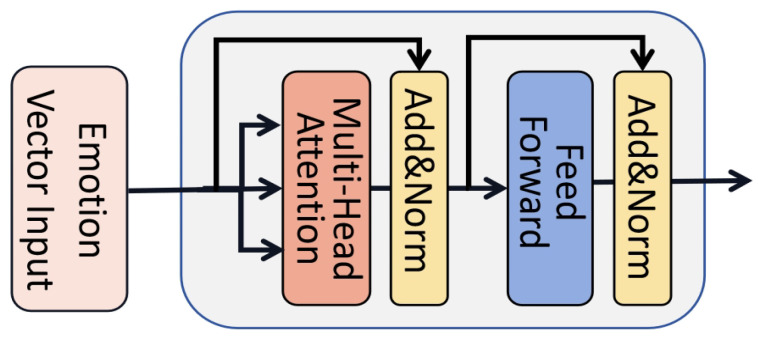
The structure of the Transformer-based temporal coding layer.

**Figure 11 brainsci-14-00344-f011:**
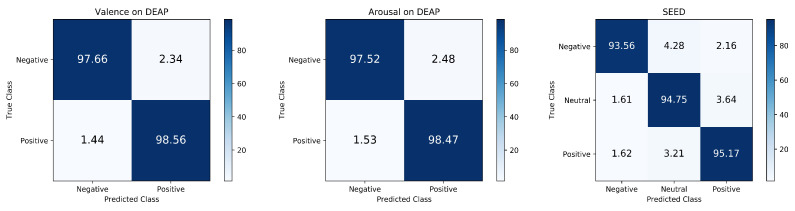
The confusion matrix of the proposed MES-CTNet on DEAP and SEED datasets.

**Figure 12 brainsci-14-00344-f012:**
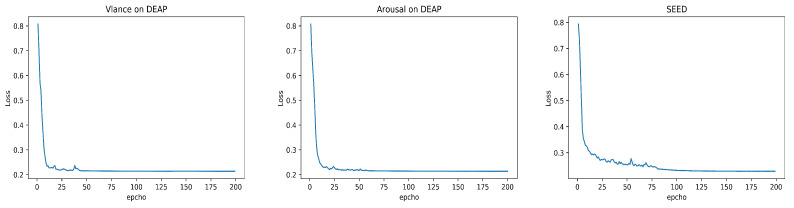
The loss map for Subject #1 of the proposed MES-CTNet on the DEAP and SEED datasets.

**Figure 13 brainsci-14-00344-f013:**
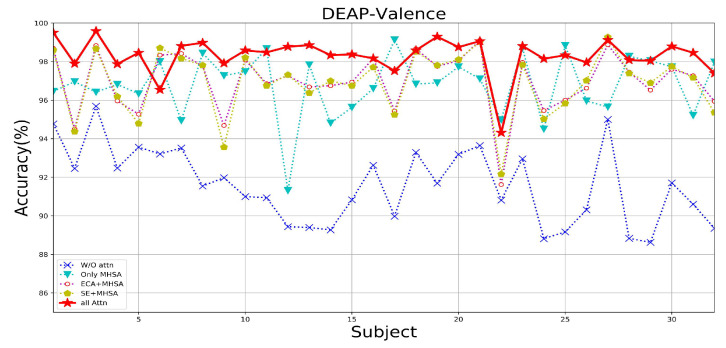
The recognition accuracy of each subject under different attention situations in the valence dimension of the DEAP dataset.

**Figure 14 brainsci-14-00344-f014:**
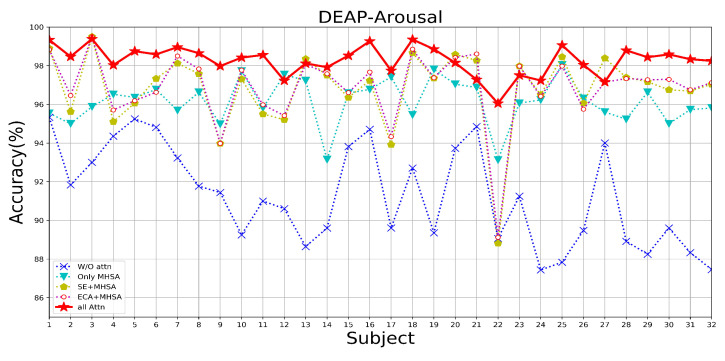
The recognition accuracy of each subject under different attention situations in the Arousal dimension of the DEAP dataset.

**Figure 15 brainsci-14-00344-f015:**
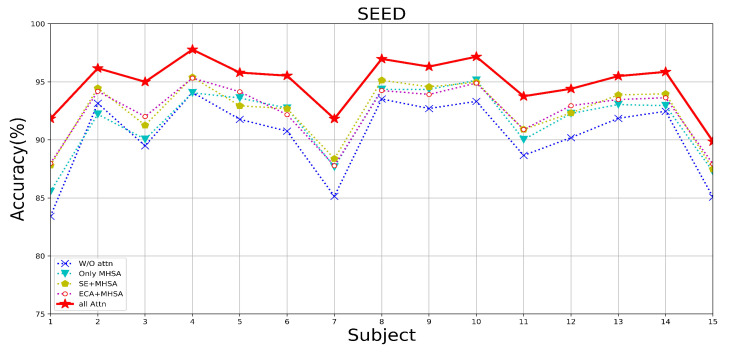
The recognition accuracy of each subject under different attention situations of SEED dataset.

**Figure 16 brainsci-14-00344-f016:**
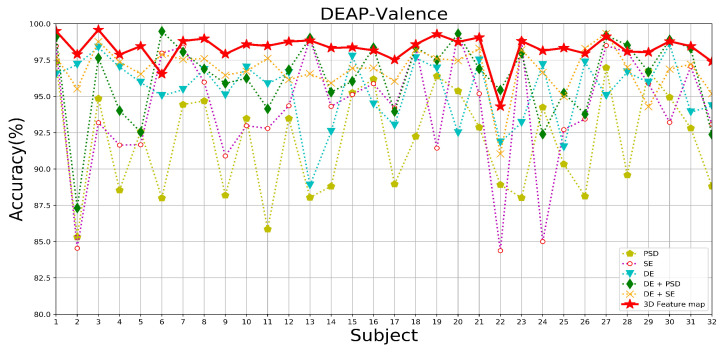
The recognition accuracy of each subject under different feature components in the valence dimension of the DEAP dataset.

**Figure 17 brainsci-14-00344-f017:**
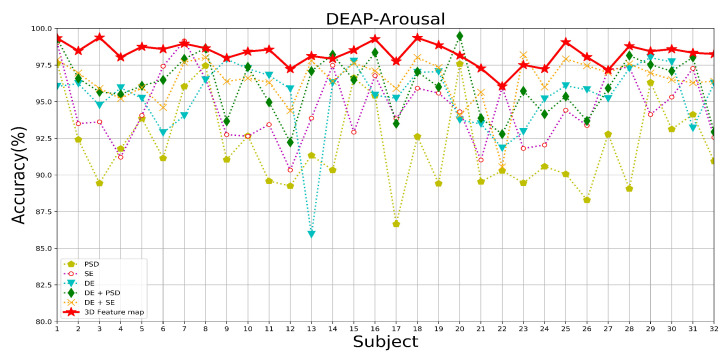
The recognition accuracy of each subject under different feature components in the arousal dimension of the DEAP dataset.

**Figure 18 brainsci-14-00344-f018:**
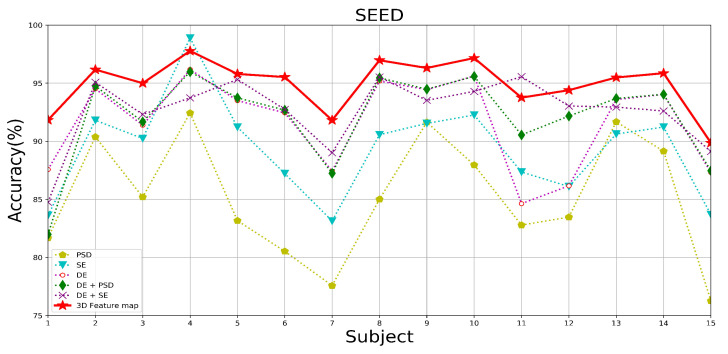
The recognition accuracy of each subject under different feature components of the SEED dataset.

**Table 1 brainsci-14-00344-t001:** Data length description.

Datase	Array	Array Sample	Array Content
DEAP	original data	32 × 32 × 40 × 8064	Subject × channels × videos × data
	3D feature map	32 × 1200 × 4	Subject × sample (3 × 16 × 18) × time
SEED	original data	15 × 3 × 62 × 47,001	Suject × session × channels × data
	3D feature map	15 × 2538 × 4	Subject × sample (3 × 16 × 18) × time

**Table 2 brainsci-14-00344-t002:** Average recall, precision, accuracy, and f1-score as well as their standard deviations (%) on DEAP and SEED dataset.

Indicator	DEAP-Valence	DEAP-Arousal	SEED
Accuracy ± std (%)	98.31 ± 0.73	98.28 ± 0.80	94.91 ± 2.10
F1-score ± std (%)	98.05 ± 1.44	97.95 ± 1.63	94.69 ± 4.67
Recall ± std (%)	98.54 ± 1.43	98.51 ± 1.53	95.16 ± 4.66
Precision ± std (%)	97.63 ± 1.46	97.45 ± 1.56	94.26 ± 4.43

**Table 3 brainsci-14-00344-t003:** Average accuracies and standard deviations (%) of different methods on DEAP and SEED datasets.

Works	Methods	Acc (%) ± Std		
		**DEAP—Valence**	**DEAP—Arousal**	**SEED**
Gao et al. [52]	CDCN	92.24 ± \	92.92 ± \	90.63 ± 4.34
Shen et al. [53]	4D-CRNN	94.22 ± 2.61	94.58 ± 3.69	94.74 ± 2.32
Tao et al. [54]	ACRNN	93.72 ± 3.21	93.38 ± 3.73	\
Deng et al. [55]	Caps-LSTM	97.17 ± \	97.34 ± \	\
Chen et al. [56]	Caps-EEGnet	96.67 ± 2.01	96.75 ± 1.90	\
This study	CTNet	96.10 ± 1.62	96.14 ± 1.85	92.41 ± 2.57
This study	MES-CTNet	98.31 ± 0.73	98.28 ± 0.80	94.91 ± 2.10

**Table 4 brainsci-14-00344-t004:** The average accuracy and F1-score values as well as their standard deviations as per the MES-CTNet model in different attention situations.

Attention Component	DEAP—Valence		DEAP—Arousal		SEED	
	**Acc ± Std (%)**	**F1 ± Std (%)**	**Acc ± Std (%)**	**F1 ± Std (%)**	**Acc ± Std (%)**	**F1 ± Std (%)**
All attention	98.31 ± 0.73	98.28 ± 1.44	98.28 ± 0.80	97.95 ± 1.63	94.91 ± 2.10	93.69 ± 4.67
ECA-MHSA	96.96 ± 1.59	96.95 ± 1.59	96.88 ± 1.89	96.87 ± 1.91	92.41 ± 2.57	92.31 ± 5.51
SE-MHSA	96.89 ± 1.66	96.89 ± 1.66	96.80 ± 2.00	96.78 ± 2.01	92.39 ± 2.69	92.11 ± 4.99
Only MHSA	96.10 ± 1.62	96.10 ± 1.62	96.14 ± 1.85	96.05 ± 1.92	91.62 ± 2.92	91.55 ± 5.36
W/O Attn	91.58 ± 1.96	89.43 ± 3.88	91.26 ± 2.56	90.33 ± 4.25	90.12 ± 3.39	89.89 ± 6.86

**Table 5 brainsci-14-00344-t005:** The average accuracy and F1-score values as well as their standard deviations as per the MES-CTNet model with different feature component inputs.

Feature Component	DEAP—Valence		DEAP—Arousal		SEED	
	**Acc ± Std (%)**	**F1 ± Std (%)**	**Acc ± Std (%)**	**F1 ± Std (%)**	**Acc ± Std (%)**	**F1 ± Std (%)**
All Feature	98.31 ± 0.73	98.28 ± 1.44	98.28 ± 0.80	97.95 ± 1.63	94.91 ± 2.10	93.69 ± 4.67
DE + SE	96.84 ± 1.54	97.02 ± 1.86	96.45 ± 1.53	96.46 ± 1.23	92.63 ± 1.96	92.25 ± 5.93
DE + PSD	96.30 ± 2.71	96.59 ± 3.75	96.12 ± 1.98	96.39 ± 4.24	92.10 ± 3.87	91.59 ± 7.12
Only DE	95.43 ± 2.28	95.42 ± 2.28	95.33 ± 2.34	95.07 ± 3.25	91.37 ± 3.88	91.37 ± 7.22
Only SE	94.25 ± 4.04	94.23 ± 4.06	94.59 ± 2.33	94.38 ± 5.16	89.28 ± 4.17	89.98 ± 8.82
Only PSD	91.88 ± 3.56	91.87 ± 3.56	92.08 ± 2.98	91.98 ± 6.61	85.41 ± 5.14	85.45 ± 5.48

## Data Availability

The database used in this study is publicly available at the following websites: DEAP— http://www.eecs.qmul.ac.uk/mmv/datasets/deap/ (accessed on 27 February 2024); SEED—https://bcmi.sjtu.edu.cn/home/seed/ (accessed on accessed on 27 February 2024).

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
