# Peer review of "MES-CTNet: A Novel Capsule Transformer Network Base on a Multi-Domain Feature Map for Electroencephalogram-Based Emotion Recognition"

_brainsci, 2024, doi:10.3390/brainsci14040344_

Round 1

Reviewer 1 Report

Comments and Suggestions for Authors

1.       In the introduction it is mentioned that “many neuroscience studies have shown that EEG signals are more closely related to emotional information than ECG signals. EDA is also one of the widely used physiological signal for emotion recognition”. Refer the current state-of-art-of art of EDA and include them in literature such as Comparison of Electrodermal Activity Signal Decomposition Techniques for Emotion Recognition and Non-Linear Signal Processing Methods for Automatic Emotion Recognition using Electrodermal Activity.

2.       Justify the choice of sampling frequency 512 Hz.

3.       The important information about the subjects is missing: No.of subjects, gender ratio, subject selection criteria, age, weight, device used for recording, ground truth, labelling, protocol etc.,

4.       Total length of the experiment is not clear.

5.       DEAP has labelled videos, can we classify the discrete emotion then?

6.       The experimental acquisition of the DEAP dataset that is similar to Fig. 10 could be generated.

7.       Instead of just Arousal and Valence classification, the four dimensions of the emotions could be classified.

8.       What is the benchmark for the choice of Electrodes.

9.       Elaborate limitations and Future scope of the study in the discussion

10.  Recent CASE dataset is using the continuous assessment of emotions. It is suggested to used CASE also compare the results.

Comments on the Quality of English Language

NA

Author Response

Dear Reviewer,

Thank you very much for providing a professional review of this manuscript, your comments and opinions are quite important for our paper, which will be very helpful for us to improve the quality of the manuscript.The detailed reply is in the attachment.

Your sincerely,

Han Ding

March 19, 2024

Reviewer 2 Report

Comments and Suggestions for Authors

In the study, researchers proposed a new Capsule Transformer Network based on Multi-Domain Feature Map for EEG-Based Emotion Recognition and tested this model on EEG signals. Emotion analysis is still an active study today, it is a field that remains up-to-date. Therefore, the study is current and appropriate to the scope of the journal. However, eliminating the following issues will increase the readability and quality of the study.

1) A flow chart of the study is not given. A flow chart should be given and the study should be visually summarized and explained.

2) The word “Ours” given in Table 5 does not sound very scientific. This word can be changed to “this study” or “this research”.

3) In unbalanced data sets, accuracy score alone is not a sufficient evaluation criterion. Therefore, calculating and interpreting values such as F1-score, recall and precision will provide more detailed information about the performance of the model. Additionally, the confusion matrix can be added and the results interpreted.

4) As mentioned in 3, accuracy score is no longer a very sufficient metric. In order to evaluate the performance of the model, balanced accuracy is now used in current studies. Researchers' calculation and interpretation of this metric will increase the quality of the study.

5) The advantages and disadvantages of the study were not mentioned. Why is the proposed method successful? Why is it more successful than other models? Does the developed model have limits? If so, what is it?

6) Loss graphs of the model are not provided. Loss graphs should be given and it should be interpreted whether the model is overfit or underfit. If there is no good training, the results obtained are of no importance.

7) Finally, the similarity report was high (27%). This rate should be at most 15%.

Comments on the Quality of English Language

Minor editing of English language required

Author Response

Dear Reviewer,

Thank you for your many professional review comments on this manuscript. Your comments and opinions are very important to the enhancement of this manuscript, and we have done our best to make changes according to your relevant suggestions, which have enriched the content of our manuscript.The detailed report is shown in the attachment.

Your sincerely,

Han Ding

March 19, 2024

Reviewer 3 Report

Comments and Suggestions for Authors

Comments on the Quality of English Language

English revision is recommended, as some sentences are incomplete and others require improvement in form. 

Author Response

Dear Reviewer,

Thank you for your careful review of this manuscript and for providing us with many valuable comments, especially on language, which were very helpful. We have checked and revised the whole text according to your relevant suggestions, and we have responded to your questions one by one.The detailed report is shown in the attachment.

Your sincerely,

Han Ding

March 19, 2024

Round 2

Reviewer 1 Report

Comments and Suggestions for Authors

Thank you very much for incorporating the suggestions, except comment #10.

CASE dataset has GSR/EDA data. Please go through the dataset once again.

Comment #10 :
Recent CASE dataset is using the continuous assessment of emotions. It is suggested to used
CASE also compare the results.
Response :
Thanks to your suggestion, we went through your introduction to the CASE dataset and
found that this is the dataset for emotion recognition via ECG. Since the primary research focus of
this report is emotion recognition via EEG signaling, emotion recognition via ECG seems to be
outside the scope of our research. However, we learned that this dataset uses a similar emotion
labeling scale as the DEAP data, which is based on the Valence-Arousal scale, and we will pay
more attention to the relationship between physiological signals similar to EDA and ECG and
emotion recognition in future studies.

Author Response

Dear Reviewer,

Thank you for your careful introduction! I am very sorry that I did not have a detailed understanding of the CASE dataset before. After your reminding, I have a more detailed understanding of the CASE dataset. The following is my understanding of the dataset :

The CASE dataset focuses on real-time continuous emotional annotations experienced by participants while watching various videos. Developed a novel, intuitive joystick-based annotation interface that allows simultaneous reporting of titer and arousal, often annotated independently. The ECG, BVP, EMG, EDA/GSR, and respiratory and skin temperature sensors were used to capture eight high quality, synchronized physiological recordings at 1000Hz and a 16-bit ADC. The subjects were 30 volunteers from different cultural backgrounds (15 males, aged 28.6 ± 4.8 years, 15 females, aged 25.7 ± 3.1 years; Age range 22-37 years). The purpose of this dataset is to induce interesting, boring, relaxing and scary emotional states through video stimulation, and collect multi-modal physiological signals in this state, including ECG, EDA/GSR, etc., through multi-modal physiological signals to produce high-quality physiological signal-based emotion recognition dataset.

In the process of understanding the data set, I developed a strong interest in this data set. Unfortunately, before starting this research, I lacked a more comprehensive understanding of other fields except EEG. Thank you for your introduction to the CASE dataset, which allows me to know more about the physiological signals of emotion recognition and understand their effectiveness. In the future research, I will expand these fields in many aspects.

Thanks again for your help, and I hope I can continue to conduct high-level research with your help.

Han

Reviewer 2 Report

Comments and Suggestions for Authors

The authors made all the revisions I suggested and updated the article.

Comments on the Quality of English Language

Minor editing of English language required

Author Response

Dear Reviewer,

Thank you for your recognition of this manuscript. Your help plays a vital role in our manuscript. We are very glad to get your recognition.

Han